# Exploring the Potential of Methanotrophs for Plant Growth Promotion in Rice Agriculture

**Jyoti A. Mohite** [1,2,†], **Kumal Khatri** [1,2,†], **Kajal Pardhi** [1,2], **Shubha S. Manvi** [1,2], **Rutuja Jadhav** [3], **Shilpa Rathod** [3] **and Monali C. Rahalkar** [1,2,*]

1   C2, MACS Agharkar Research Institute, G.G. Agarkar Road, Pune 411004, Maharashtra, India; jyotimohite@aripune.org (J.A.M.); kumal.microbio@gmail.com (K.K.); kajalpardhi@aripune.org (K.P.); shubhamanvi@aripune.org (S.S.M.)
2   Department of Microbiology, Savitribai Phule Pune University, Ganeshkhind Road, Pune 411007, Maharashtra, India
3   Biotechnology Department, Rajarshi Shahu Mahavidyalaya Latur, Latur 423511, Maharashtra, India; rutujadhav009@gmail.com (R.J.); shilparathod821228@gmail.com (S.R.)
*   Correspondence: monalirahalkar@aripune.org; Tel.: +91-20-25325119
†   These authors contributed equally to this work.

**Abstract:** Rice fields are one of the important anthropogenic sources of methane emissions. Methanotrophs dwelling near the rice roots and at the oxic–anoxic interface of paddy fields can oxidize a large fraction of the generated methane and are therefore considered to be important. Nitrogen fixation in rice root-associated methanotrophs is well known. Our aim in this study was to explore the potential of methanotrophs as bio-inoculants for rice and the studies were performed in pot experiments in monsoon. Ten indigenously isolated methanotrophs were used belonging to eight diverse genera of Type Ia, Type Ib, and Type II methanotrophs, including the newly described genera and/or species, *Methylocucumis oryzae* and *Methylolobus aquaticus*, as well as *Ca.* Methylobacter oryzae and *Ca.* Methylobacter coli. Additionally, two consortia (*Methylomonas* strains and *Methylocystis-Methylosinus* strains) were used. Nitrogen fixation pathways or *nifH* genes were detected in all of the used methanotrophs. Plant growth promotion (PGPR) was seen in terms of increased plant height and grain yield. Nine out of twelve (seven single strains and two consortia) showed positive effects on grain yield (6–38%). The highest increase in grain yield was seen after inoculation with *Ca.* Methylobacter coli (38%) followed by *Methylomonas* consortium (35%) and *Methylocucumis oryzae* (31%). *Methylomagnum ishizawai* inoculated plants showed the highest plant height. *Methylocucumis oryzae* inoculated plants showed early flowering, grain formation, and grain maturation (~17–18 days earlier). In all the pot experiments, minimal quantities of nitrogen fertilizer were used with no additional organic fertilizer inputs. The present study demonstrated the possibility of developing methanotrophs as bio-inoculants for rice agriculture, which would promote plant growth under low inputs of nitrogenous fertilizers. Although the effect of methanotrophs on methane mitigation is still under investigation, their application to reduce methane emissions from rice fields could be an added advantage.

**Keywords:** methanotrophs; rice agriculture; bio-inoculants; plant growth promotion; *nifH* gene; Type I methanotrophs; Type II methanotrophs

## 1. Introduction

In Asia, paddy cultivation is a major agricultural practice as rice constitutes a staple food source for a large proportion of the human population [1]. Rice farming poses two major environmental and financial challenges: a. the need for nitrogenous fertilizers like urea, which, besides being expensive, are also environmentally hazardous due to nitrous oxide emissions and ammonia volatilization [2], and b. *methane emissions* from rice fields as a result of anaerobic processes. As rice paddies are cultivated in a water-logged

environment, they become a source of high methane, which is the second most important greenhouse gas [1,3,4]. Methane has a 26 times greater capacity to absorb heat than carbon dioxide ($CO_2$) (IPCC, 2013). It is estimated that rice paddies account for ~10% of global methane emissions [4]. It is hypothesized that the methane emissions from rice paddies would otherwise be at least twice as high if the anaerobically produced methane was not consumed by methanotrophs [5].

In paddy fields, niche differentiation and aerobic methane oxidation using methanotrophs are fueled by oxygen diffusion via rice plants and water tables. In such an ecosystem with a high potential for $CH_4$ production, it is estimated that 40–90% of the methane produced could be oxidized to $CO_2$ by methanotrophs before it is emitted to the atmosphere [1], and references within. Thus, methanotrophs act as key biofilters in wetland ecosystems including rice fields.

Methanotrophs utilize $CH_4$ as a source of carbon and energy by oxidizing it with molecular oxygen [6]. The proteobacterial aerobic methanotrophs belong to three families: *Methylococcaceae*, *Methylocystaceae,* and *Beijerinkiaceae*. The family *Methylococcaceae* (Type I methanotrophs) includes the genera *Methylobacter, Methylomonas, Methylomicrobium, Methylococcus*, *Methylocaldum*, *Methylocucumis*, *Methylomagnum*, etc., whereas the family *Methylocystaceae* (Type II methanotrophs) includes the genera *Methylosinus* and *Methylocystis* [7]. Physiological differences between the two families are correlated with different ecological preferences [8].

Many of the aerobic methanotrophs that can fix atmospheric nitrogen and both type I and type II methanotrophs are known to have nitrogen fixation pathways [9,10]. Such nitrogen fixation pathways have also been recently detected in the genomes of *Methylocucumis* sp., *Methylobacter* sp., *Methylolobus* sp., and *Methylomagnum* sp. of methanotrophs [11–16]. Sessitsch et al. in 2012 found the presence of the *nifH* gene for *Methylocystis* sp. in the rice metatranscriptome [17]. Methanotrophs are estimated to fix around $1.2 \times 10^{14}$ g N/yr globally, thus making a significant contribution to global nitrogen fixation [18]. Recent stable isotope probing experiments showed that methanotrophy highly contributes to biological nitrogen fixation [19]. This property of nitrogen fixation in methanotrophs could be explored for plant growth promotion in rice agriculture, thus opening a new class of bioinoculants. In the present study, we studied the potential of methanotrophs as future bio-inoculants for rice agriculture in terms of promoting the growth of rice plants. A minimal amount of nitrogen fertilizer was used to check the growth-promoting effect of methanotrophs. The increased abundance of methanotrophs may also lead to a reduction in methane emissions, which could be advantageous for reducing global warming.

## 2. Materials and Methods

### 2.1. Methanotrophs Used and Genome Analysis for the Detection of Plant Growth Promotion Genes/Pathways

Ten pure cultures representing eight genera of methanotrophs from Type I (from *Gammproteobacteria*) and Type II (from *Alphaproteobacteria*) were used in this study. These cultures were isolated by us as a result of around ten years of work on the isolation and cultivation of methanotrophs. These ten cultures were as follows: *Methylomagnum ishizawai* strain KRF4 [16], *Methylolobus aquaticus* FWC3 [14], *Methylocucumis oryzae* strain BM10 [11,20,21], *Ca.* Methylobacter coli strain BlB1 [15], *Ca.* Methylobacter oryzae KRF1 [12,13], *Methylomonas* sp. strain Kb3 [22,23], *Methylomonas sp.* strain WWC4 (Gammaproteobacterial/Type I methanotrophs), *Methylocystis* spp. SnCys [24], *Methylosinus sporium* strain KRF6, and *Methylosinus trichosporium* strain KRF10 (*Alphaproteobacterial*/Type II methanotrophs). The draft genomes of most of these cultures, except for *Methylosinus trichosporium* strain KRF10 and *Methylomagnum ishizawai* strain KRF4, were deposited by us in NCBI and the accession numbers are listed in (Table 1). As *Methylomagnum ishizawai* KRF4, *Methylosinus* KRF10 were 99–100% similar based on the 16S rRNA gene to the type strains of *M. ishizawai* and *M. trichosporium*; we used the genome information of the type strains/ other available strains in these two cases for the analysis. The DNA extraction, genome sequencing, and analysis

were performed as described (Supplementary Methods). The genomes were analyzed for the presence of different types of plant growth promotion (PGP) genes or pathways for nitrogen fixation, production of indole acetic acid production (IAA), phosphate solubilization, cellulose degradation, and trehalose synthesis.

**Table 1.** Summary of methanotrophs used in this study.

| Name of the Organism | Representative Strain | | | Genome NCBI Number | Identification Using *pmoA* Gene | | Identification Using 16S rRNA Gene | |
|---|---|---|---|---|---|---|---|---|
| | Strain Name | GeneBank Accession Number | | | Nearest Match (with Type Strain) | % Similarity | Nearest Match (with Type Strain) | % Similarity |
| | | *pmoA* Gene | 16S rRNA Gene | | | | | |
| *Methylocucumis oryzae* | BM10 | MT366581 | MN462841 | LAJX01 * | *Methylococcaceae bacterium* Sn10-6 | 100 | *Methylococcaceae bacterium* Sn10-6 | 100 |
| *Ca.*Methylobacter coli | BlB1 | MH424899.2 | JADMKV01 | JADMKV01 | *Methylobacter marinus* A45 | 97.57 | *Methylobacter marinus* A45 | 98.5 |
| *Methylocystis* spp. | Sn-Cys | KT156638.1 | MZ562889.1 | JAERVJ01 | *Methylocystis iwaonis* SS37A-Re | 99.34 | *Methylocystis iwaonis* SS37A-Re | 99.93 |
| *Methylomonas* spp. | Kb3 | KP862532 | KM995837 | PIZT01 | *Methylomonas denitrificans* FJG1 | 95.72 | *Methylomonas denitrificans* strain FJG1 | 99.13 |
| *Methylomonas* spp. | WWC4 | MH806338.1 | MH64454.1 | JAATWI01 | *Methylomonas methanica* S1 | 93 | *Methylomonas koyamae* FwR12E-Y | 97 |
| *Ca.* Methylobacter oryzae | KRF1 | MH806336.1 | MK511847.1 | RYFG02 | *Methylobacter tundripaludum* SV96 | 90 | *Methylobacter tundripaludum* SV96 | 98.6 |
| *Methylolobus aquaticus* | FWC3 | MH806335.1 | MH7895511 | SEYW01 | *Methylocaldum marinum* S8 | 85.9 | *Methylocaldum marinum* S8 | 94 |
| *Methylosinus sporium* | KRF6 | WP_216281891.1 | MZ562999.1 | JAHLJF0.1 | *Methylosinus sporium* NCIMB 11126 | 91 | *Methylosinus sporium* NCIMB 11126 | 98.43 |

* Reference genome of *Methylocucumis oryzae* Sn10-6 was used.

Methanotrophs from the genera *Methylomonas*, *Methylocystis*, and *Methylosinus* were most commonly isolated from rice rhizospheres in Indian rice fields in our earlier studies [25]. Several strains of these genera were available with us, and hence, these were also shortlisted for the plant growth promotion experiments. The *Methylomonas* strains screened were BM6, KRF3, IS1, and DMS2; the *Methylosinus* strains screened were KRF7, KRF8, KRF9, and KRF10; and the *Methylocystis* strain screened was SnCys.

The nitrogen fixation abilities of most of the strains have already been documented earlier by their growth in a nitrogen-free NMS medium under micro-aerophilic conditions [11,12,14,15,20]. Similar experiments were set up for all the other strains used and not tested for growth on nitrogen-free media under micro-aerophilic conditions with methane as the substrate (*Methylomonas* strains, *Methylocystis* and *Methylosinus* strains, and *Methylomagnum* strain KRF4) (Supplementary Methods). Similarly, *nifH* gene was amplified for these strains using the primers *nifH* 1 and 2 [26] described in detail in Supplementary Methods. The nitrogen metabolism pathways of the selected strains of methanotrophs were constructed using the KEGG web server. The cultures that showed nitrogen fixation pathways and/or other plant growth promotion genes in their genomes were used for further pot experiments.

### 2.2. Pot Experiments Using Methanotrophs as Bioinoculants

#### 2.2.1. Trial with Pure Cultures of Type I and Type II Methanotrophs

A first trial for the pot experiments using ten pure cultures of methanotrophs as liquid bioinoculant was taken in 2021. Based on the presence of plant growth promotion pathways and nitrogen fixation genes, pure cultures of type I and type II methanotrophs, *Methylomagnum ishizawai* strain KRF4, *Methylolobus aquaticus* FWC3, *Methylocucumis oryzae* strain BM10, *Ca.* Methylobacter coli strain BLB1, *Ca.Methylobacter oryzae* KRF1, *Methylomonas* strain Kb3, *Methylomonas sedimenticola* strain WWC4, *Methylocystis* spp. SnCys, *Methylosinus sporium* strain KRF6, and *Methylosinus trichosporium* strain KRF10 were used for pot experiments

to assess their plant growth promotion in rice plants. Two sets of controls without inoculum were maintained, where only sterile inorganic medium was added to the pot. All the methanotroph cultures (Figure 1, Supplementary Figure S1) used in this study were maintained on solid- and liquid-modified NMS media [25,27], followed by incubation at 28 °C in the presence of methane: air mixture (20:80). The headspace gases were replaced periodically.

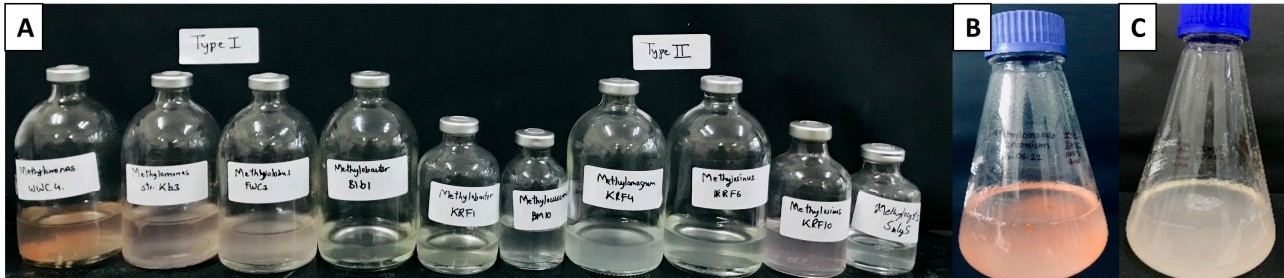

**Figure 1.** Methanotrophs used in the trials for rice plant growth promotion experiment. (**A**) Methanotrophs inoculum used in the 2021 rice plant growth promotion trial, (**B**) defined consortia of type I *Methylomonas* strains used in the 2022 rice plant growth promotion trial, and (**C**) defined consortia of type II *Methylocystis-Methylosinus* strains used in the 2022 rice plant growth promotion trial.

All experimental trials were taken in the rice growing season (monsoon) between June and October. Garden soil from a local nursery in Pune was purchased and used for all experiments. Plastic pots (height 35 cm, diameter 40 cm) were filled with 25 kg of red soil (purchased from a local nursery) and flooded with rainwater for seven days. Twenty-five-day-old rice plants of the Indrayani variety were purchased from a local rice farmer in the Mawal area, near Kamshet, and were immediately transplanted in each pot (16 seedlings/pot). Four rice hills with 4 seedlings per rice hill and four rice hills per pot were planted. Each pot was then fertilized with 0.5 g of Sufala® fertilizer that contained NPK (15:15:15). All the pots were flooded for the next four days after transplanting. Out of twelve pots, ten pots were used to test the effect of methanotrophic inoculums, and two pots were kept as controls (without treatment of any methanotrophs). After five days of transplantation of rice seedlings, a 300 mL inoculum (0.2–0.3 OD$_{600}$) of pure strains of methanotrophs was added to the respective pots. Two control pots were added with 300 mL of sterile NMS medium. All the pots were kept in the institute's open space and received natural sunlight and rain. As this was one of the first trials of using methanotrophs as microbial bio-inoculants and the cultures grew only up to ~0.2–0.3 OD in most of the cases, we added maximal volume of the inoculum (~300 mL). The water levels were maintained at ~5 cm in each pot, excess rainwater was removed, and occasionally, if the rainfall was less, tap water was added to keep the plants flooded. After the flowering stage, flooding was not maintained, and the plants were only watered if the soil dried up (to mimic the conditions in the rice fields). The experiments continued till the grain maturation stage and dried grains were collected.

### 2.2.2. Trial with Mixed Cultures

In a second pot experiment trial in 2022, two defined consortia/mixed cultures consisting of type I methanotrophs *Methylomonas* strains (BM6, KRF3, IS1, and DMS2), and type II methanotrophs *Methylosinus* and *Methylocystis* strains (KRF7, KRF8, KRF9, KRF10, and SnCys) were prepared to check their rice plant growth potential. The defined consortium was made by mixing equal amounts of each culture grown to ~0.2–0.3 OD$_{600}$ (Figure 1, Supplementary Figure S1). All the methanotroph cultures used in this study were maintained on a solid NMS media as well as in a liquid NMS medium, followed by incubation at 28 °C in the presence of methane: air mixture (20:80). The headspace gases were replaced periodically.

A similar pot experiment as above was set up in the year 2022, except that instead of adding individual cultures, we used defined consortia of *Methylomonas* and *Methylocystis-Methylosinus* for checking the plant growth promotion. In the year 2021, we saw that methanotroph single cultures showed plant growth promotion; however, a large single dose was applied. In 2022, we reduced the dose of the inoculum to 1/10th, and ~25 mL of each consortium was added in three consecutive doses. The first dose of the inoculum was added on the fifth day of the transplantation of rice seedlings. A 25 mL consortium with 0.2–0.3 OD (2–3 $\times$ $10^8$ cells/mL) was added to the respective pots. This was repeated after every ten days and a total of three doses were given to each pot. In addition to 0.5 g of Sufala® fertilizer that contains NPK (15:15:15), one gram of urea was added to the transplanted plants to each pot to maintain a total basal nitrogen level (~52.5 kg N/ ha). Usually, ~150 kg N/ha is added to Indian rice fields (https://agritech.tnau.ac.in/agriculture/agri_nutrientmgt_rice.html), accessed on 24 August 2023. Two sets of controls without any inoculation were added with a sterile NMS medium (25 mL) per dose. All the other conditions were maintained as described above with the pots exposed to open sunlight and rain.

For both trials, the results were noted by measuring individual plant length, percent increase in plant height compared to control, days of a flowering stage, grain yield, and percent increase in grain yield compared to control (Table 1). The experiments continued till the grain maturation stage and dried grains were collected.

## 3. Results and Discussion

### 3.1. Confirmation of Nitrogen Fixation Potential in Methanotrophs

Rhizospheric bacteria play a key role in biological nitrogen fixation, providing vital nutrients for plants. Co-evolution has led to these interactions, making PGPR-based approaches promising for ecological agriculture and increased productivity in crops like rice.

Plant growth promotion properties in a bacterium facilitate the plant nutrient uptake from the surrounding environment [28–30]. Nitrogen is the most essential and important nutrient for plant growth [2]. Therefore, it becomes a major growth-limiting factor in agriculture [2]. Rice agriculture needs a large quantity of nitrogenous fertilizers in the form of urea or other ammonia-based fertilizers. At present, rice cultivation globally consumes more than 10 million tons of nitrogen fertilizer [2]. India spent $5.1 billion to import 8.1 million tonnes of urea during 2022–2023. Non-symbiotic biological nitrogen fixers such as *Azosprillum* can contribute to a significant amount of nitrogen up to 10 kg N/ha in cereals [30]. Currently, the available bio-inoculants for rice are limited to *Azotobacter and Azospirillum,* e.g., BioNit G^R by Kanbiosys, India. The bio-inoculation of the rice fields with *Azolla* fern, which has a symbiotic association with *Anabaena* and fixes the atmospheric nitrogen, is also used. Despite the availability of these bio-inoculants, farmers have to use nitrogenous fertilizers for rice agriculture. To reduce this dependency, there is a need for research into new bio-inoculants for rice [2].

Methanotrophs, especially Type I methanotrophs, are active near rice roots [31]. A symbiosis of methanotrophs with rice roots has been suggested earlier, where methanotrophs dwelling near roots receive energy and carbon from methane and fix nitrogen delivering to the plants [32]. Thus, methanotrophs can be efficient bio-inoculants in terms of their nitrogen fixation and methane oxidation abilities and are worth exploring.

In the current study, methanotrophs, which were mostly isolated from rice fields in India, were used to explore their potential as novel bio-inoculants to boost rice agriculture. In the initial screening, it was clear that nitrogen fixation pathways were present in all of the eight strains for which we performed genome sequencing (Supplementary Table S1). All the other strains used in this study, strains used in the consortia and *Methylomagnum ishizawai* strain KRF4, showed a positive *nifH* gene amplification (Supplementary Table S2, Supplementary Figure S2). Though the type strain of *Methylomagnum ishizawai* RS11D-Pr does not possess nitrogenase genes, other strains, e.g., *Methylomagnum ishizawai* strain 175 isolated from the Philippines, showed a nitrogenase operon and growth in nitrogen-free

medium [33]. The 16S rRNA gene of strain KRF4 showed 99.6% similarity to that of strain 175; additionally, the cell size and cell shape of strain KRF4 were similar to those of strain 175. Similar to strain 175 [33], *Methylomagnum ishizawai* strain KRF4 showed growth in a nitrogen-free medium with methane as the C source. All methanotrophs were able to grow in a nitrogen-free mineral medium under micro-aerophilic conditions.

The classical genes for plant growth promotion, such as IAA production, 1-aminocyclopropane-1-carboxylate-ACC deaminase, etc. [30], were absent in the genomes of the methanotrophs. Under salinity/osmolarity conditions, rhizosphere-associated methanotrophs can accumulate compatible solutes such as trehalose [30]. Trehalose production is beneficial for plants to fight drought conditions or water scarcity. The capacity to produce trehalose was also seen in the genomic information of a few of the studied methanotrophs (Supplementary Table S1). A few other plant growth promotion genes, e.g., cellulose degradation, which helps the plant to establish itself in the rhizosphere area, and a few genes for phosphate solubilization were detected in some of the methanotrophs (Supplementary Table S1). To summarize, the ability to fix nitrogen was most commonly seen in all the methanotrophs, and hence, pot experiments were performed to study their efficacy in plant growth promotion of rice plants.

### 3.2. Effect of Methanotrophic Bioinoculant on Rice Plant Growth

In the study, methanotrophs were used to make bio-inoculum and were tested for their effect on the growth of the Indrayani rice variety, the most popular rice variety in Western Maharashtra, majorly grown in the Mawal, Mulshi, and Donje regions. Rice is normally grown by transplanting rice seedlings that are 25–30 days of age. Rice seeds are grown by farmers in small patches as nurseries on rice fields and after 25–30 days of seedling age, these are transplanted to the main field, which is puddled with rainwater. This variety takes about 140 days to complete its cycle after sowing, and the average grain weight of a thousand seeds is ~16 g [34].

All the pot experiments in this study were performed in the rice growing season, i.e., monsoon season (June to October), and the rice plant seedlings ~25 days after sowing were brought from local farmers in both trials. This would ensure that the seedlings were healthy and uniform, as our earlier attempts to use seedlings grown in pots were not completely successful. We purposefully used very low amounts of nitrogen fertilizer: 7.5 kg N per ha in the first trial, and 52.5 kg N per ha in the second trial, which is 1/20th and 1/3 times, respectively, of the regular dose used for rice in India, which is ~150 kg N/ha (https://agritech.tnau.ac.in/agriculture/agri_nutrientmgt_rice.html), accessed on 24 August 2023. Low nitrogen has been key to triggering nitrogen fixation in grasses such as switchgrass [35]. We hypothesized that the low nitrogen fertilizer input would trigger nitrogen fixation in methanotrophs. Both the pot trials performed in the year 2021 (using individual strains) and 2022 (using consortia) were successful in terms of obtaining filled grains and completing the entire plant life cycle.

In the first pot trial, single methanotroph cultures were used in a single high dose (300 mL of 0.2–0.3 OD culture) after transplantation, and we observed that except for one culture (*Methylosinus* strain KRF6), all other cultures showed growth promotion in terms of enhanced plant growth and/or enhanced grain yield (Figure 2, Table 2). An increase in plant height compared to control pots was observed in methanotrophs inoculated with *Methylomagnum ishizawai*. This was followed by *Methylocucumis oryzae and Ca.* Methylobacter coli inoculants, and the *Methylomonas* strains Kb3 and WWC4 (Table 2). Increased plant height is considered to be an important parameter in terms of plant growth promotion.

We also observed early flowering in *Methylocucumis oryzae* inoculated rice plants where the rice plants flowered ~17 days earlier compared to the control plants (Table 1). Due to the early flowering, the grain formation and grain maturation also occurred earlier with *Methylocucumis oryzae* inoculated plants, whereas in the rest of the inoculated plants, the flowering stage was seen 3–7 days earlier or with the control plants (Table 1).

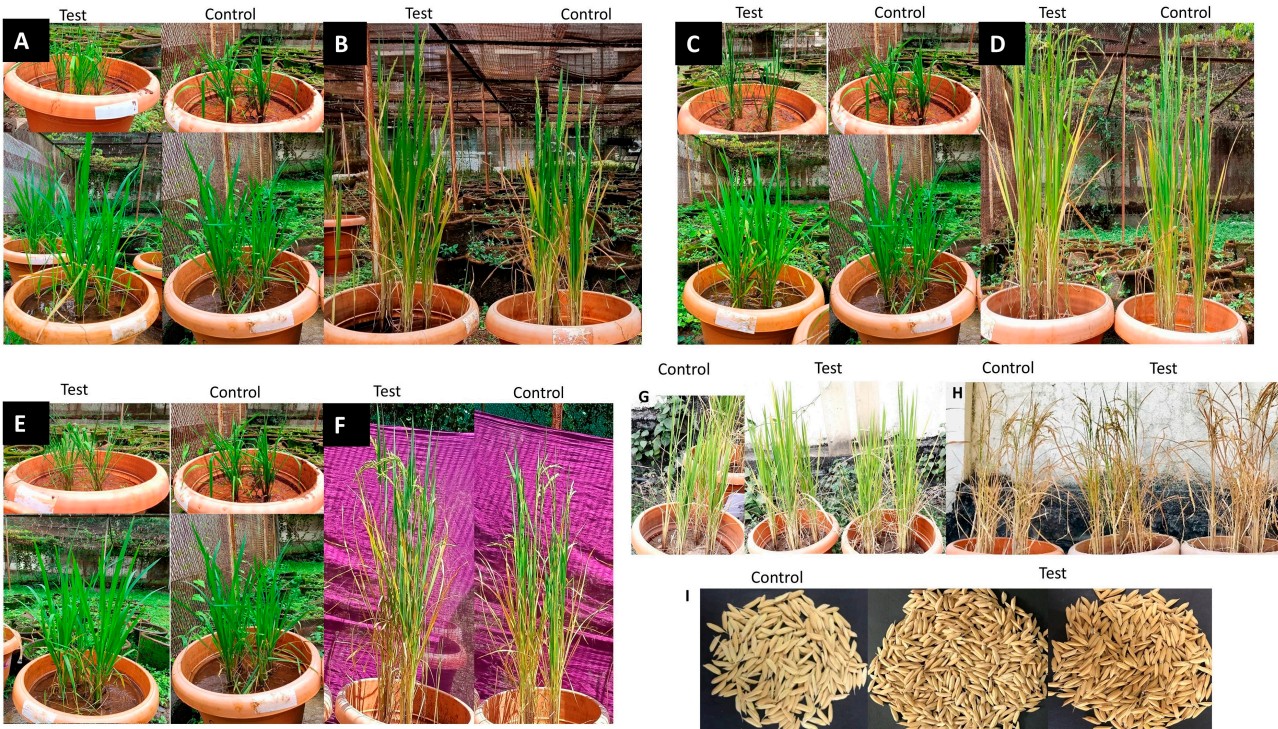

**Figure 2.** Pot experiment results in 2021. (**A**) Plants inoculated with *Methylomagnum ishizawai* strain KRF4 at the transplantation stage and (**B**) at the maturation stage; (**C**) plants inoculated with *Methylocucumis oryzae* strain BM10 at the transplantation stage and (**D**) at the maturation stage (early flowering was observed compared to control and all other methanotrophic strains); (**E**) plants inoculated with *Ca.* Methylobacter coli strain BlB1 at the transplantation stage and (**F**) at the maturation stage. Pot experiment results in 2022. (**G**) Rice plants with bio-inoculum of type I *Methylomonas* consortium and type II *Methylocystis* and *Methylosinus* consortium at the growing stage; (**H**) rice plants with bio-inoculum of type I *Methylomonas* consortium and type II *Methylocystis* and *Methylosinus* consortium at the maturation stage; (**I**) Matured and dried grains of rice inoculated with no inoculum (control), *Methylomonas* consortium, and *Methylocystis*- *Methylosinus* consortium (L to R).

The grain yield of methanotrophs inoculated rice plants was more in most of the cases than the control plants. A considerable increase in grain yield was seen in methanotrophs-inoculated rice plants. *Ca.* Methylobacter coli strain BlB1 inoculated Indrayani plants showed the highest increase in grain yield, which was 38%, the highest among all inoculated plants (Table 1), followed by *Methylomonas* consortium (35%) and *Methylocucumis oryzae* (31%) compared to the control plant yield. *Methylomagnum ishizawai* and *Methylomonas* Kb3 showed a ~13% increase in grain yield compared to that of the control plants. Thus, the Type I methanotrophs showed a better effect in terms of plant growth promotion. *Methylolobus aquaticus* strain FWC3 and *Methylocystis* strain SnCys showed increased plant height but no effect on grain yield to uninoculated control plants (Table 1). *Methylosinus sporium* strain KRF6 showed no increase in plant height and a little decrease in the grain yield compared to the control plants.

In the second trial, the type I consortium of *Methylomonas* strains and type II consortium of *Methylosinus* and *Methylocystis* strains showed positive effects on plant growth in terms of both plant height and grain yield (Figure 2, Table 2). *Methylomonas* consortia showed a considerable increase in grain yield (35%), and *Methylocystis-Methylosinus* consortia showed a ~12% increase in grain yield. Overall, both the methanotrophic bio inocula positively affected the growth of Indrayani rice plants, but further studies with different plant varieties and combinations of methanotrophs are needed to strengthen the results and make them usable in the field.

**Table 2.** Growth parameters comparison of Indrayani rice plants grown in the presence of methanotrophs v/s the control.

| Sr. No. | Strain Name | Mean Plant Height (cm) | % Increase in Plant Height Compared to Control | Flowering Stage (Day after Sowing) | Total Yield (Weight of the Grains (g)) | % Increase in Total Yield Compared to Control |
|---|---|---|---|---|---|---|
| 1 | *Ca.* Methylobacter coli strain BlB1 | 88 ± 3 | 12 | 110 | 22 | 38 |
| 2 | *Methylocucumis oryzae* strain BM10 | 88 ± 3 | 12 | 97 | 21 | 31 |
| 3 | *Methylomagnum ishizawai* strain KRF4 | 91 ± 3 | 16 | 106 | 18 | 13 |
| 4 | *Methylomonas* sp. Kb3 | 87 ± 7 | 12 | 111 | 18 | 13 |
| 5 | *Methylosinus trichosporium* strain KRF10 | 85 ± 7 | 12 | 107 | 17 | 6 |
| 6 | *Methylomonas* sp. WWC4 | 85 ± 7 | 12 | 107 | 17 | 6 |
| 7 | *Ca.* Methylobacter oryzae strain KRF1 | 84 ± 3 | 8 | 111 | 17 | 6 |
| 8 | *Methylolobus aquaticus* strain FWC3 | 83 ± 7 | 8 | 111 | 16 | - |
| 9 | *Methylosinus sporium* strain KRF6 | 77 ± 4 | - | 107 | 13 | - |
| 10 | *Methylocystis* sp. Sn-Cys | 88 ± 1 | 12 | 107 | 15 | - |
| 11 | Control pots 2021 (mean) | 78 ± 3 | - | 113 | 16 | - |
| 12 | Type I *Methylomonas* consortia | 78 ± 2 | 13 | 114 | 23 | 35 |
| 13 | Type II *Methylosinus-Methylocystis* consortia | 75 ± 2 | 9 | 114 | 19 | 12 |
| 14 | Control pots 2022 (mean) | 69 ± 2 | - | 113 | 17 | - |

All of the Type I methanotrophs were more successful compared to Type II methanotrophs, i.e., *Ca.* Methylobacter coli [27], *Methylomonas* consortium, *Methylocucumis oryzae* [11,20], *Methylomonas* Kb3 [22], and *Methylomagnum ishizawai*. All of these cultures were isolated from rice fields, except for *Ca.* Methylobacter coli. *Ca.* Methylobacter coli was isolated from black buck feces, with black buck being an herbivore [27]. In the first pot trial, the effect of individual strains was understood. In the second trial, we chose three genera: *Methylomonas*, *Methylocystis* and *Methylosinus*. The strains from these three genera were isolated very frequently from Indian rice fields [25]. These cultures are relatively easy and fast to grow, and hence, they were used in a consortium where the strains were highly related to each other; it was hypothesized that they do not interact in an antagonistic way. *Methylomonas* was also identified to be the dominant nitrogen fixing bacterium in several studies, including SIP experiment [19].

Methanotrophs are usually present in large numbers near rice rhizospheres [31,36,37], and nitrogen fixation genes from methanotrophs are amongst the transcribed *nifH* genes in rice roots [17]. Methanotrophs are known to contribute to plant nitrogen in rice fields, especially under low nitrogen fertilizer conditions [32]. Rice agriculture needs excessive urea or ammonium-based fertilizers; hence, there is a need for new bio-inoculants for rice agriculture, which would be helpful in achieving nitrogen fixation and mineralization. In a recent study, it was shown that the nitrogen fixation activity was triggered by methanotrophs [19].

DNA-based Stable Isotope Probing (SIP) indicated that gamma proteobacterial *Methylomonas* like methanotrophs dominated nitrogen fixation in methane-consuming roots [19]. The $^{15}$N experiments suggested that about 42.5% of the fixed nitrogen was available in the form of $^{15}$N labeled ammonium. This study highlighted the importance of roots associated with methanotrophs as both the biofilters of methane and microbial engines of bio-available nitrogen for rice growth.

Our study is one of the first studies where pure cultures and two defined consortia of methanotrophs were used to check their effects on rice plant growth. Most of the methanotrophs used in the current study were able to fix atmospheric nitrogen and showed the presence of complete nitrogen fixation pathways in their genomes. Hence, these methanotrophs were most likely able to affect plant growth by making more nitrogen available via nitrogen fixation. Though we have not directly shown that the increase in the grain yield or height of the plant was due to the nitrogen fixation by the methanotrophs, this could be the subject of future studies.

In our study, we performed a bio-augmentation strategy by providing additional methanotrophs as bio-inoculants and then checked for their plant growth promotion effects. Though our study was performed at the pot level, there was a clear positive effect on the growth of the inoculated plants versus un-inoculated control plants. Further studies in the field in a systematic manner would help us understand the efficacy of using methanotrophs for plant growth promotion in rice and open a new arena of bio-inoculants for rice agriculture, which is the need of the hour.

**Supplementary Materials:** The following supporting information can be downloaded at: https://www.mdpi.com/article/10.3390/methane2040024/s1.

**Author Contributions:** Conceptualization: M.C.R.; Methodology: J.A.M., K.K., K.P., S.S.M., R.J. and S.R.; Formal analysis and investigation: J.A.M., K.K., K.P. and M.C.R.; Writing—initial draft preparation: M.C.R., K.K. and J.A.M.; Writing—review and editing: M.C.R., J.A.M. and S.S.M.; Funding acquisition: M.C.R.; Supervision: M.C.R. All authors have read and agreed to the published version of the manuscript.

**Funding:** This study was supported by the Department of Science and Technology, SERB grants: (EMR/2017/002817), and POWER fellowship grant (SPF/2022/000045) provided to MCR. Authors KK and KP are thankful to Council for Scientific and Industrial Research and University Grant Commission, respectively, for the research fellowships. Similarly, JAM acknowledges the SARTHI program for the junior research fellowship. SSM acknowledges the SERB project: CRG/2021/000941 for providing a junior research fellowship.

**Institutional Review Board Statement:** Not applicable.

**Informed Consent Statement:** Not applicable.

**Data Availability Statement:** All the sequence data have been submitted to NCBI and the genome accession numbers, 16S rRNA gene sequence accession numbers are provided in the tables.

**Acknowledgments:** We would like to thank Pranitha Pandit for the help during measurements in the growth of the plant height and for taking photographs during the pot trials. We would also like to thank the rice farmers for providing us with the rice seedlings.

**Conflicts of Interest:** The authors have no relevant financial or non-financial interests to disclose.

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
