# Peer review of "Exploring the Potential of Methanotrophs for Plant Growth Promotion in Rice Agriculture"

_methane, doi:10.3390/methane2040024_

Round 1

Reviewer 1 Report

In this study, the author added pure or mixed cultures of various methane oxidizing bacteria to rice culture pots and found that these strains can promote plant growth and increase yield. This is an interesting and promising research that is worth publishing. However, there are still some issues with the manuscript, as listed below.

1. I have some questions about the setting of the control. In the method, 300 ml of methanotrophic bioinoculant was added to the treatment pots, while nothing was added to the control pots. The better growth of rice plant in the treatments may be due to the added nutrients in 300 ml of medium and the nutrients brought in by the methanotrophic cells. It is more reasonable to set up another control set, in which an equal amount of sterilized methanotrophic bioinoculant is added, and each treatment corresponds to a control to confirm that the better growth of rice plant in the treatment group is indeed due to the addition of live methane oxidizing bacteria. The author should explain these possibilities in the discussion section. For example, can the nutrients brought in by medium and cells be ignored?

2. Line 97-98, What is the gas composition when the strain is cultured in nitrogen free medium? Is the bottle kept stationary or shaken? The results mentioned that all strains can grow under nitrogen free conditions, so how much can the OD value be reached and how long does it take? Please supplement the corresponding results in the form of a table or figure.

3. Line 137-147, There is a significant overlap between this section and the previous content, which needs to rewrite.

4. Based on your results, is there a difference in the promotion effect of type I and type II on rice? What are the possible reasons?

5. Nitrogenase coding genes were not detected in genome of Methylomagnum ishizawai KRF4, but it can grow in nitrogen-free mineral medium. Please explain.

6. Line 224-235, Only the nitrogen fixation function of methanotrophs has been discussed here. It is also necessary to discuss other growth promoting functions of methanotrophs and whether they may have played a role in this experiment.

Other minor comments:

1. Line 59, you missed members of Beijerinckiaceae. Also the genera of type I you list here are not complete.

2. Line 67, “sp.” should not be italic.

3. Line 92 and 172, Methylosinus should be italic.

4. Line 97, full name of IAA?

5. Line 107”emthanotrophs".

6. Line 171, full name of ACC?

7. Line 209, Methylocystis should be italic.

8. Line 340, rewrite this legend. For example, Rice plant at transplantation stage that inoculated Methylomagnum ishizawai strain KRF4.

9. Supplementary figure 2. Add a bar to indicate the cell size.

Reviewer 2 Report

The manuscript by Mohite et al. describes a set of mesocosm experiments to determine the potential use of methanotrophs for bioaugmentation.  The main highlight of the study is the demonstration of improved grain yield in rice plants following the addition of methanotroph inocula.

The study is well-written and in general requires only minor revisions to be suitable for publication. There are a few areas where some clarification of methods and the reasoning behind them is needed.

Methods, lines 85-97: Please add some details about genome sequencing and annotation.

Methods, line 121-130, 142-153: Was the rainwater stored (and if so, some details are needed), or just left outside? If so, were measurements kept of how much precipitation there was each day – if there was little or no rain, were pots topped up to keep them continuously flooded?

Methods, lines 148-150: clarification is needed on the method (see below comment), and some explanation for why trials a and b were set up differently

Methods/Results: Why was the concentration of fertiliser increased in the second trial?

Methods/Results: Why were these particular strains chosen for consortia? Some were not tested in trial a (e.g. Methylomonas BM6, KRF3, IS1, DMS2) or if tested, were not the most efficient plant-growth promoters – why not use M. coli in a consortium?

Supplementary Figure 1 – I don’t think this is a useful addition – nif operons are stated in Supplementary Table 1, and there’s no reference to other nitrogen pathways in the manuscript.

Supplementary Figure 2 isn’t referred to in the text, and could also be omitted.

Specific comments:

Line 19: Please change to “Many methanotrophs...”

Line 23: “Urea” should be “urea”

Line 24: “of the strains” should be added after “the genomes”

Line 26, 30: This should be Ca. Methylobacter

Line 46: prevalent gas in what?

Line 48: This should be “methane emissions”

Line 50: Please change to “methane was not consumed...”

Line 50: “at” is not necessary

Line 56: “Thus supporting the fact that” is not necessary

Line 68: Please add “and” before Methylomagnum, and remove “genus”

Line 69: ‘for” should be “from”

Line 74: “there” should be “this”

Line 92: “mentioned” should be “listed in”

Line 93: Please italicise strain names

Line 102: Please add “(three strains)” after Methylosinus sporium

Line 105: This should refer to Supplementary Table 1

Line 107: Should be “methanotrophs”

Lines 110-114: Strain names can be abbreviated form this point, as in trial b

Lines 116-117: “a” is not necessary before media types

Line 121: Is the weight correct? It doesn’t seem to match the pot volume stated

Lines 140-148: Could be reduced to “ as above”

Lines 148-150: I don’t understand this sentence – were trial b plants transplanted three times into different pots, have three inoculations of methanotrophs and/or fertilised three times? What is the “active tillering stage”? Why was this different to the set up in trial a?

Line 170: “The classical genes...” needs a reference

Line 172: Please italicise strain names

Line 175: This should refer to Supplementary Table 1

Line 183: Move comment about weight to e.g. line 202

Lines 206, 208: “ ~ ” is not necessary

Line 220: From Table 2, the Methylomonas consortium should be listed before Methylocucumis

Line 229: “this by” is not necessary

Line 233: Should have a reference at the end (18 again?)

Table 2 – The final two columns have different spacing so the rows don’t line up

Supplementary Table 2 – The final strain is listed as Methylosinus iwaonis...

Minor editing of English language required
